# Plotting the Words of Econophysics

**DOI:** 10.3390/e23080944

**Published:** 2021-07-23

**Authors:** Gianfranco Tusset

**Affiliations:** Department of Economics and Management, University of Padua, via del Santo 33, 35123 Padua, Italy; gianfranco.tusset@unipd.it

**Keywords:** lexical evolution of econophysics, text as data, correspondence analysis

## Abstract

Text mining is applied to 510 articles on econophysics to reconstruct the lexical evolution of the discipline from 1999 to 2020. The analysis of the relative frequency of the words used in the articles and their “visualization” allow us to draw some conclusions about the evolution of the discipline. The traditional areas of research, financial markets and distribution of wealth, remain central, but they are flanked by other strands of research—production, currencies, networks—which broaden the discipline by pushing towards a dialectical application of traditional concepts and tools drawn from statistical physics.

## 1. Introduction

The introduction in physics of a new kind of statistical law, or, better, simply a probabilistic law, which is hidden under the customary statistical laws, forces us to reconsider the basis of the analogy with the […] statistical social laws. It is indisputable that the statistical character of social laws derives, at least in part from the manner in which the conditions for phenomena are defined. It is a generic manner, i.e., strictly statistical, allowing countless complexes of different concrete possibilities. On the other hand, […] we are induced to ask ourselves whether there also exists here a real analogy with social facts, which are described with a somewhat similar language (p. 258) [1].

These words were written by a great theoretical physicist, Ettore Majorana, as a preamble to an article, *The Value of Statistical Laws in Physics and the Social Sciences*, on the convergence of natural and social sciences that Majorana wrote around 1930 before disappearing in 1938.

Majorana was hoping that physics and social sciences (including economics) would move in the direction of a shared language. If the social sciences, economics in particular, had always looked to classical physics as a model of scientific rigor, Majorana wanted the new physics and social sciences to converge on a common statistical field. 

Majorana’s message introduces the short journey we are about to make in the discipline of econophysics, that more than others have taken up the invitation to develop a research area in which natural and social sciences converge. Although there have been episodes that have anticipated some of its contents—from the far Bachelier random walk (1900) [2] and Pareto Law (1896–1897) [3] to the more recent Farjoun, and Machover *Laws of Chaos* (1983) [4], to name just a few—econophysics was born in the early nineties of the last century, with the celebrated article by Nunzio Mantegna on the *Lévy walks* (1991) [5]. Therefore, it has thirty years, maybe few to understand if it has been able to collect and develop Majorana’s message, but enough for the definition of its own disciplinary identity.

Econophysics is a broad and magmatic field in terms of content and methods, as is well highlighted by at least a dozen highly scientific texts that deal in detail with the statistical, mathematical and theoretical facets of this new field. To understand if econophysics is moving in the direction desired by Majorana, if indeed that common language is on the horizon, we will consider the scientific articles on econophysics published during these years, analyzing them from a linguistic point of view, aware that words mean contents, methods, objectives.

The encounter between natural sciences and social sciences raises a theme that cannot be ignored and that goes beyond the very search for a common language: it is the theme of laws. In the world of relationships between individuals, of human behavior, are there social and economic laws that can be compared to the invariant laws that characterize the natural world? Econophysics does not ignore the problem, indeed it has made it a topic of discussion.

The linguistic reconstruction of econophysics will therefore be an opportunity to understand how positions are evolving on this point, to understand if the search for laws that characterizes physics represents a dominant feature also in the activity of econophysicists.

Section 2 presents the literature and our research methods. Section 3 illustrates the frequency of the main lexical cluster words identified in the texts considered as shining light on the evolution of the econophysics lexical corpus. Section 4 focuses on the possible correlation between the identified lexical clusters. Section 5 is devoted to the visual representation and analysis of econophysics words. Section 6 contains some concluding remarks.

## 2. Literature, Methods, and Results

Econophysics has known various moments in which it has discussed itself. The debate on empirical regularities that took place between the two components, economists and physicists, in 2006 [6,7], should be mentioned, as well as prolonged research on individual theoretical and methodological aspects of the discipline [8,9,10,11,12,13]. Also articles that periodically take stock of the state of econophysics should also not be ignored [14,15].

To try to understand the directions that econophysics is taking, we reconstructed its lexical development over the period from 1999 to 2020. The technique is that of “text as data” to define the frequency matrix of words used in econophysics articles. This matrix is then used for the realization of a scatterplot as a picture of the evolution of the lexicon of econophysics.

The approach to “text as data” proposed here can be defined as “bags of words” [16], i.e., the texts are broken down into words and short combinations of words (single or in segments of two or three), whose frequency is counted in relation to the year or quarter of publication, the latter considered ‘active variables’. Therefore, the words and segments constitute the ‘tokens’, placed in a row of a large matrix that in each column presents the active variables chosen, in our case quarters and whole years. Thus, words and the text segment become the starting point of our analysis. In short, the words and segments that populate text are statistically analyzed to identify meta-trends and meta-behaviors that would not otherwise be immediately apparent.

For construction of the matrix and, thus, the scatterplot, 510 econophysics articles published between the years 1999 and 2020 were used, mainly in *Physica A* (287 or 56%), partly in *The European Physical Journal B* (165, 32%), and a minority (58, 12%) from other journals (*Physical Review E*, *Contemporary Physics* and few others). In the article only the words were used: therefore, the mathematical or statistical content is not taken into account.

Of course, this is only a fraction of the econophysics articles that have appeared since 1991. Nor would it be materially possible to analyze all the articles (each article must be cleaned to be included in the overall corpus). In selecting the articles, we used the following criteria.

First, we relied on the search engines of the sites of the two area journals considered *Phys. A* and *Eur. Phys. J. B*, which have been attentive to econophysics since the early days of its appearance. Rather than moving on the basis of a definition of econophysics, we relied on existing internal classifications.

For each of these journals, the choice of articles is proportional to the distribution of econophysics articles that have appeared in the various years (e.g., the 287 articles in *Phys. A* should be representative of the 1616 articles that the *Phys. A* website identifies as econophysics articles for the same period). The distribution of articles by subareas (the clusters) reflects the distribution of topics among the articles in the year examined.

The priority given to these two journals (*Phys. A* and *Eur. Phys. J. B*) stems from their emphasis on econophysics. Few articles published in other journals were included because they were particularly significant, often for the insights into the significance of the discipline they contained.

The impossibility, for now, of constructing a large corpus on the basis of most of the articles on econophysics published in journals of different subject areas limits the interpretative scope of an analysis constructed on the frequency of words. We believe, however, that, although not complete, the sample used here is sufficient to have a first significant result of the relative distribution of words and, therefore, of the sub-areas of research, during the period considered.

Words are certainly among the protagonists of our story, an idea that can be schematized in three steps. First, we treated the words and segments contained in the articles like our data, while the period of publication represents the variable under which words and segments are grouped. Thus, the early step was to construct a large matrix containing the frequencies of the overall words and articulated by quarter/year. The matrix or contingency tables contains 6915 rows (words/segments) and 88 columns (quarters from 1999 to 2020). All grammatical terms of 2 and 3 syllables and all words with fewer than 6 occurrences were removed from the corpus.

The analysis of the words contained in the matrix allows us to extract some lexical clusters that facilitate the modeling of the evolution of the econophysics lexical corpus. FIN* includes all words/segments concerning financial topics; DIST* the same for the broad area of distribution of wealth, income and other variables, often an object of sociophysics analysis. “Power law” is not included because considered more a tool than an object, as could be income or wealth; PROD* includes words/segments referring to the industrial and production world; CURR* refers to words concerning any kind of currency circulation, including cryptocurrency; and NETW* including all words concerning networks and complex networks.

To investigate the attitude towards the search for invariant laws, we introduced two other lexical clusters, STAT* and NONST*, which include words/segments related, the first one, to contents proper to statistical physics (“power law”, “multifractality”, “stationarity” are included here), more properly macro that do not imply the analysis of individual choices of agents; the second, to an analysis of ‘rumors’, of non-stationarity, of specificities often evident on the micro level (“minority game”, “agent-based”, “reflexivity”, “non-stationarity” and so on) and emphasizing potential “noise”, “instability” and similar phenomena.

The second step is to regress the time series organized into quarters from 1999 to 2020 regarding the lexical clusters above to determine the extent to which they are treated in isolation or together and how the STAT*-NONST* relationship of the debate is integrated into the treatment of other content. One can rightly question the use of VAR regression for time series regarding words, but it is more than an exact measure of potential causality, here we are interested in identifying trends to guide us in our treatment of the large topics above. The quantitative analysis is functional to the qualitative analysis developed in the next step.

Finally, the third step is dedicated to visualization and analysis of the words over the period considered here. We decided to adopt correspondence analysis as the most appropriate analytical method to visualize the words/segments contained in the papers in relation to the above active variables. This is an exploratory data processing technique belonging to multivariate statistics and designed to analyze the above matrixes containing frequencies, that is, measures of correspondence between rows and columns. Correspondence analysis was well suited to our purpose because our study lacked an *a priori* hypothesis to verify; it enabled us to identify systematic relationships between variables, without any prior expectations regarding the nature of these relationships [17,18].

Scatterplots showing the outcomes of this linguistic analysis are grounded on relative frequencies. Axes of the scatterplot were selected according to the level of inertia, i.e., the variance exhibited by the *active variables*. In other words, the active variables (in this case, whole years) were arranged according to the variance characterizing their own lexicon. The two pairs of active variables with the greatest distance in their lexicon identified the horizontal and vertical axes. Then we could also work with illustrative or *case variables*, i.e., words belonging to rows that can be pinpointed on the scatterplot showing the distribution of the dataset, and then associated with the active variables. This step helps to clarify the characteristics of the lexicon used by econophysicists.

The multiplicity of the active variables generates the multidimensionality of the data matrix. Exploratory factorial analysis enables this multidimensionality to be reduced by transforming data into noncorrelated variables and building factorial or semantic axes that constitute “points of view” on the phenomenon observed (p. 62) [19]. These points of view are contextual in that they display relationships across a broad corpus of texts by reducing the amount of information. Specific software is needed to analyze such a large dataset, so we used Automatic Lexical and Textual Processing for the Analysis of Content (*TALTAC*) and *R* to manage the corpus (both led to similar matrices), and *SPAD* to extract the figures relating to our study.

Briefly, the results of this lexical analysis. Econophysics tends to gradually widen its field of application, extending it to an increasing number of economic and social phenomena. This is a process which undoubtedly broadens the sphere of influence as well as the competence of the discipline. This process, however, pushes towards a dialectical, not dogmatic, application of the principles inherited from statistical physics: suffice it to mention the universality of the laws or the invariance of scale. This dialectical process, more common to the social sciences than to the natural sciences, does not weaken econophysics, on the contrary, it makes it more dynamic and alive. However, its application implies a challenge for econophysics, which remains, or aspires to remain, a natural science.

## 3. Measuring Lexical Clusters

To obtain a preliminary viewpoint, we reconstructed trends in the relative frequency of the five clusters. Figure 1 shows the results of this calculation in quarters since 1999. Since data are relative frequencies, the number of articles per quarter was used to calculate the total number of words on which to calculate the relative frequency of those words of interest to us. Clearly, we referred to the average number of words per article. Thus, each frequency is calculated on the total number of words appearing in the articles considered in that quarter.

Taking a quick look at Figure 1, what stands out is the prominence in terms of the relative frequency of the lexical cluster FIN* and partially of DIST*.

In particular, FIN*, which includes words referring to options, stocks, and all financial products, represents a constant in the interest of econophysicists but, contrary to what one might imagine when thinking about the financial origins of econophysics, it becomes dominant, from the perspective of lexical frequencies, from 2012 onwards, reaching various peaks, those of highest intensity in 2015, 2017 and 2019. DIST*, distribution of wealth and income, represents, since the early years of the period considered here, an important topic in the research of econophysicists, characterized by some peaks in different periods (the highest in the third quarter of 2008, and smaller ones in 2005, 2007, 2014 and 2020 respectively).

The trend of the PROD* cluster, including the reference to the real economy, is interesting. The attraction of econophysicists to industrial and production issues, without presenting relevant peaks, appears to present greater strength since mid-2014. As we will see later, these are the years in which interest in networks, financial and otherwise, grows.

The CURR* topic only exploded after 2014 and later, when cryptocurrency became the subject of analysis by econophysicists. Often treated as a financial asset, cryptocurrencies are also of interest as a means of circulation, an aspect that has prompted us to keep them separate from financial securities. Also included in this topic are all words that refer to monetary circulation, a recurring theme in the treatises on econophysics.

Although it indicates an approach rather than an area of economic/financial activity, we have also included here network, NETW*, whose prominence has grown, especially after the first years of the last decade, to the point of becoming an autonomous research area with respect to econophysics, an aspect that also explains its decreased frequency among the words of the discipline after 2015.

Stationarity or nonstationarity as well? To try to reconstruct the prevailing orientation among econophysicists, we have reported in Figure 2 the trend of two lexical clusters expressing the two possibilities. In the STAT* cluster we find those words/segments that indicate a preference for an econophysics faithful to physical statistics that analyzes the behavior of aggregates independently of that of individuals, searches for power laws and scale-invariance. The NONST* cluster, on the other hand, takes into account the development in the direction of nonstationarity, scale-dependence, reflexivity, behavior of individual agents not just aggregates, including agent-based computation.

The figure shows that econophysics is not solely the discipline of statistical physics devoted to aggregates. Interest in the two directions coexists showing various peaks of NONST* as well as STAT* peaks.

## 4. Correlation between Lexical Clusters

Is econophysics a discipline that deals primarily with financial markets? Or does it touch on a wide range of aspects of economic and financial life? Can relative frequencies tell us anything about the relationship between the topics (lexical clusters) that are the subject of econophysics work? Once the main lexical clusters were defined, we tried to study their evolution from 1999 to 2020. As a first step, we tried to understand whether the various research strands have, over time, constituted a single disciplinary corpus or have remained substantially separate, also in light of the debate on the macro or micro-orientation of the discipline. The idea is to test the existence of causality and correlations between lexical clusters represented by time series related to word frequencies.

To seek such hypothetical causalities or correlations, we start by testing Granger causality between the available time series: FIN*, DISTR*, PROD*, CURR* and NETW*. Adopting a level of confidence of 5 percent, we have identified the following outcomes (see Table A1 in Appendix A):

FIN*, DIST*, PROD* and NETW* are lexical clusters that have no causality or correlation between them. These sub-areas grow, expand, but within hypothetical sub-disciplinary boundaries. Thus, the lack of correlation between them should be read.

Both DIST* and NETW* affect CURR*, which is equivalent to saying that the debate on currency circulation is influenced by the debates on distribution and production, mainly that on distribution, if we consider the two *p*-values (Table A1). As the CURR lexical cluster is the most recent in terms of development, the causality of which it is the subject is, perhaps, symptomatic of a lesser stiffening or closure of these sub-areas.Finally, the debate over NETW* is affected by the debate over STAT*.

The interesting fact is that FIN*, DIST*, PROD*, and NETW* represent a world unto themselves, not talking to each other or being influenced by other debates.

The two lexical clusters STAT* and NONST* concerning the more the approach than the content yielded the following outcomes.

The STAT* orientation is conditioned by the FIN*, DIST*, NETW* and by the same NONST* cluster.The NONST* orientation does not affect any cluster, but is influenced by FIN*, DIST*, and NETW*.

The latter two causalities feed into the dialectical process above. Causality concerns not only STAT, which descends from statistical physics, but also NONST*, which instead challenges it. Hence the intertwining of the two lexical clusters, observable in Figure 2.

Finally, considering STAT* and NONST* in isolation, one cannot ignore the Granger causality from the latter to the former. Indeed, together with the observation of the absence of autocorrelation in the time series of the two variables, such causality induces an interpretation of this type: the centrality of statistical physics, power law and scale invariance need to be frequently reaffirmed in the face of the doubts evoked by NONST* words/segments.

These (few) causal relationships are only statistical hypotheses, however, and need to be validated. Consistently with the approach of our work, we opt for a textual validation: rather than seeing whether the above statistical hypotheses can be refuted or not, we use these hypotheses as a key to interpret how econophysics’ lexicon changes over time.

## 5. Visualizing the Words of Econophysics

Recalling the English adage that ‘a picture is worth a thousand words’, our analysis of the texts produced during the crisis can be enriched by taking a further step and moving from number to image (image of words, in this case). The scatterplots presented here “can be regarded as maps” of the use of words and segments concerning topics in the econophysics corpus. The scatterplot simply “communicates [...] information” (p. 5) [17]. Correspondence analysis provides “ways for describing data, interpreting data, and generating hypotheses” without a theoretical model or preconceived hypothesis.

How can we interpret the scatterplot obtained by correspondence analysis? If a given word/segment is close to an active variable (a given year, for instance), this means that it characterizes speeches or discussion papers published at the time. On the other hand, words/segments that are common to most or all active variables (years) considered are to be found in the center (centroid) of the figure.

The scatterplot is constructed using a second matrix that differs from the one used for above figures solely because of the active variables (columns), the first quarters (88) and now years (22). In contrast, the words/segments (rows) remain the same (6915).

Our word/segment cloud lies in a *c* − 1 dimensional space, where *c* is the number of active variables, the 22 years in our case. The choice of coordinates to be represented is such as to ensure the widest representation of words/segments consistent with their distance (in row and column) from the mean profiles located in the center of the plane. In short, the widest linguistic variability is guaranteed.

If *i* = 1, ..., *r* are the words/segments considered here, *j* = 1, ..., *c* the active variables i.e., the years analyzed, *n* the total of words/segments occurrences, *n_i_*_._ the total of the matrix *i*-row, *n*_.j_ the total of the matrix *j*-column, we can express the distance, *d*, between two words/segments *i* and *i’* as Pearson chi-square distance (*χ*^2^) in the form:
(1)$$d^{2}\left(i,i^{\prime }\right)=\overset{c}{\underset{j=1}{\sum }}\frac{n}{n_{.j}}{\left(\frac{n_{ij}}{n_{i.}}-\frac{n_{i^{\prime }j}}{n_{i^{\prime }.}}\right)}^{2}\;\;$$

The Euclidean distance weighting in Equation (1) results in a reassessment of the low-frequency components and a scaling of the high-frequency components. The very low frequencies (less than 6 occurrences) were removed to prevent them from weighing too heavily in the distance calculations due to the weighting (p. 107) [19].

Briefly, the scatterplot in Figure 3 shows the evolution of the vocabulary of econophysics articles. On the axes we find the inertia, which can be considered as an index of lexical change: the higher its value, the higher the variability of the words contained in the analyzed texts. In our case, it is quite low on both axes: 10.02 and 6.81 percent. This means that this representation explains only 16.83 percent of the total variability. By changing the combination of axes, we get lower values of total inertia. This result can be interpreted by stating that, in these twenty-two years, the vocabulary of econophysicists has changed little and very gradually. The movement can be read clockwise. Arranging the years in a sufficiently orderly sequence shows that the change has been gradual, but continuous. We will focus on the gradually introduced changes in the lexicon of econophysics, but the low variance makes it clear that previously used words and segments continue to be used. In other words, as new concepts are entered, the previous ones were retained. The most common words or segments, such as “Brownian motion”, “statistical physics,” “power law,” found around the origins of the axes, are not shown because they were shared by most of the articles.

The lexicon used in econophysics in the period under consideration follows a sort of clockwise trajectory that goes from the left side of the axis to the fourth quadrant on the lower right, passing through the second and first quadrants. To make its interpretation easier, the lexical path has been divided into five phases, each of which is lexically characterized by marking a stage in the construction of the vocabulary of econophysics. The titles attributed to each phase look more to marginal novelty than to the main body of scholarship from that period, reiterating the interest in change at the margin.

Briefly, reviewing the five phases will help us understand if and how the topics at the heart of econophysics have changed and how the orientation towards STAT* and NONST* has changed over time. Remember that words/segments are positioned in relation to years is done based on their respective relative frequencies, calculated on the set of words used in each year in the articles considered between 1999 and 2020.

About the distribution of words/segments in general, we can observe how it is rather spherical in the first four phases, signifying a rather weak inertia, while it shows a dilation in the fifth phase (2018–2020), proof that in the last years here considered the lexicon tends to show evident and not only gradual signs of change. The years 2019 and 2020 contribute 18.3 and 12.8 percent, respectively, to the formation of the horizontal axis and 12.2 and 31.3 percent, respectively, to the formation of the vertical axis. There are not many words/segments that characterize the fifth phase, but they show considerable weight in structuring the entire word/segment distribution.

### 5.1. Phase I—Statistical Aggregates

The first phase, subarea I of Figure 3, corresponds to the first six years, from 1999 to 2004. During this period, the new research area was presumably reinforcing its methodological and conceptual pillars drawn from statistical physics: “gases”, “Brownian motion”, “option pricing” and “Lévy distribution” testify that we are in the world of statistical physics applied mainly to financial markets. The idea that the theoretical properties of gases could be extended to a market composed of many agents, each operating as a particle, aroused great interest. The goods exchanged could be of any kind, including the income distributed throughout the economy as a whole. During this early phase, the discipline’s focus is primarily on the outcome of the many unpredictable exchanges that occur in a market, not on what causes them or on the decision-making process of the agents. Statistical econophysics shows more interest in “predictions” of future prices or rather future price changes than in understanding how the market works. The direct challenge to economic theory anchored in individualism and the role of the representative agent is plain.

The aggregates of statistical physics produce distributions that cannot ignore what Mandelbrot [20,21] has shown, namely, that the distribution function of asset prices deviates significantly from Gaussian. Part of the subsequent development of econophysics is the result of this debate, including the need to normalize “stationary distributions.”

About distribution, “Power law” appears in this first phase, but within econophysics, “power law” is something of a focal point that has allowed and contributed to the discipline’s ability to stay within the confines of its inheritance from physics. Perhaps it is inappropriate to talk about power law science [22], but its popularity stems from the common belief that “small occurrences are extremely common, while large occurrences are extremely rare.”

Taking a brief look at the stances taken on power law, can well represent the opinion of the early physicists engaging in socio-economic research: “Physicists are often fascinated by power laws. The reason for this is that complex, collective phenomena do give rise to power laws which are *universal*, that is, to a large degree independent of the microscopic details of the phenomenon. The power laws emerge from collective action and transcend individual specificities” (p. 105) [23]. However, power law models “contain multiplicative noise” and “lead to nonuniversal exponents that depend on the value of the parameters”. It thus becomes necessary to model observations at the microscopic level to explain the decay of volatility correlations on this level (p. 112) [23]. Agent-based microscopic models were still advocated by Ausloos et al., for the same purpose, i.e., to determine “scaling exponents and universal laws” (p. 2) [24]. However, “although [power law] is probably not the universal law that some have claimed it to be, it is certainly a powerful and intriguing concept that potentially has applications to a variety of natural and man-made systems.” (p. 346) [25].

Single agents, however, do not disappear in the aggregates of statistical physics, even in this first phase focused on statistical sets. The word “agents” itself weighs in at a significant 0.2 and 0.6 percent in determining the horizontal and vertical, respectively (the 100 percent is obtained by summing the individual contributions of the 6915 words/segments to the formation of the horizontal and vertical axes). Various types of noise can make their appearance in the study of aggregate phenomena or distributions. Typically, these noises are related to microscopic analyses of how markets work. This explains the presence in the first subarea of segments such as “minority game”, a variant of Brian Arthur’s El Farol bar problem [26], and “minority group”, typical of computational models based on agents making decisions based on their memory of what happened in the past. Agent-based analysis, which has developed independently, is thus gradually being drawn into the galaxy of econophysics, given the need to explain microphenomena. Agent-based analysis has also generated an abundant literature on models based on assumptions very different from those that characterize statistical aggregates.

A sort of dialectic between macro and micro, between scale invariance and multiscale, between stationarity and non-stationarity makes its appearance since the first phase originating that causation between NONST* and STAT* mentioned above.

### 5.2. Phase II—Stationary or Nonstationary Processes?

Econophysics was a discipline that reached maturity in a few years. As Figure 3 shows, the period from 2005 to 2009 appears, in fact, characterized by those words that define its identity: “stylized facts”, “Pareto law”, “wealth distribution” and so on. Consistent with DIST’s peak of the years 2006–2008 visible in Figure 1, “distribution” becomes a key word in econophysics, identifying an area of research, the distribution of wealth, that has begun to represent a specific field in the discipline. Since those years, it has been possible to state that wealth/income distribution analysis and financial market research have represented, not without overlap, the two main areas of research in the discipline.

When talking about income distribution and price changes, the notion (crucial for econophysics) of “stylized facts” is quite common. Simply put, these are phenomena that are primarily visible at the meso and macro levels, and usually lack a micro theoretical foundation. A stylized fact allows generalization without reference to time or spatial contextualization. Although the notion of “stylized fact” is widely accepted by statistical physicists aiming to explain aggregate or macro phenomena, it remains shrouded in a kind of vagueness, perhaps a legacy of its economic origin. Some recognized and universal stylized facts—such as distribution laws, option pricing and risk control—sit alongside less accepted stylized facts, such as trends in GDP or inflation. However, stylized facts also remain central to their use in the study of financial markets [27].

However, even at a stage when econophysics recognizes its roots in statistical physics, it does not fail to discuss them, thus making this discipline a living field of research.

It is a fact that deviations of price time series from random walk behavior and “price distribution” have been studied, moving also in the direction of stylized self-organizing facts. “Self-organizing” and “self-organization” together with “group of agents” highlight the novelty of this phase. A system characterized by self-organizing criticality is able to move towards a stable critical regime that is characterized by long-range correlations and free-scale power laws. From an economic perspective, we can look at the ability of markets to organize themselves by means of intermediate actors, such as groups of firms or sectors, or even uncoordinated agents [28]. If markets are able to converge toward stability, there is no need to analyze their internal or micro dynamics.

The transition from the microstate to the macrostate level or “phase transition” is part of the analysis of markets and socio-economic systems. “Self-organization” has a role in any phase transition. In 2007, Newman wrote: “There has been much excitement about self-organized criticality as a possible generic mechanism for explaining where power-law distributions come from […] Self-organized critical models have been put forward not only for forest fires, but for earthquakes, solar flares, biological evolution, avalanches and many other phenomena” (p. 347) [25].

But self-organization does not necessarily mean homogeneity of agents. The models postulated the distinction between inactive agents (“chartists”) and active agents (“fundamentalists”), and the feedback between price fluctuations and the number of active agents, implicitly admitting that agents can decide whether or not to enter the financial market based on their “predictions” regarding price changes. The choice involves a price dynamic that does not guarantee that the probability distribution will remain stationary over time. On the contrary, there may be a “nonstationary distribution” (p. 386) [29].

Not only that. The evolution of the income distribution debate has involved the assumption that agents have “saving propensities” [30] or saving parameters [31], which affect the volume of exchange between agents, viewed as particles colliding to exchange energy. When saving is allowed, the intensity of this exchange decreases, and the distribution consequently takes on a new shape (p. 166 ff) [32].

The ability of markets to organize themselves in a stable manner has been discussed. “Thermodynamics,” which appears in the previous step, is connected to “stationarity” and “non-stationarity.” The latter reminds us of what McCauley wrote in that very year: “There is no reliable analogue of energy in economics, and there are very good reasons why no meaningful thermodynamic analogy can be constructed” [7]. Thermodynamic equilibrium would require a stationary equilibrium, whereas markets and production are not stationary, nor are increases in the time series, with the consequence that growth processes can be understood by considering not only their variation over time, but also their initial conditions.

Time matters. With respect to financial markets, “non-stationarity” in time series could be caused by secular trends or other long-term factors that do not permanently characterize the observed phenomenon. In other words, the parameters of a process or distribution may change. This aspect distinguishes economics from physics. Clearly, nonstationary processes force us to set aside the ergodic condition and to reconsider “non-ergodicity” as a norm in economic processes (p. 3180) [33]. Are there concepts from physics that cannot be applied to economics? However, the parallel between natural and social sciences, rich in both similarities and differences, continues to be at the heart of econophysics.

### 5.3. Phase III—Zero-Intelligent Agents

We know that aggregates gave rise to empirical events because of so many causes that it was impossible to explain them by adopting a deductive approach. Phenomena were the product of too many causes to be investigated. Decision-making processes were ignored. However, are the interacting agents/particles that animate these phenomena incapable of making decisions? Are they zero intelligence [10,34]? Zero intelligence, the lexical protagonist of the third phase (2010–2012), must be conceived referring not to the decision-making capacity of agents, but to the inability to link the global outcome under observation to the behavior of the underlying microstructures. Agents are random factors, therefore assumptions about their behavior are not necessary to obtain stylized facts. The direction seems diametrically opposed to that of perfect rationality.

While minority game models have been proposed primarily to explain some stability and stationarity weaknesses at the aggregate level, zero-intelligence units are introduced into agent-based computation to assert “implicit microfoundations”: individuals represent “black boxes” that are sources of unpredictable noise subject to objective constraints. Usually, microfoundations are explicit because the choice (optimization) mechanism is fully specified and functions as an essential explanatory factor. Here, agents are efficient even if their rationality is not explicit. What matters is the macro phenomenon, regardless of any individual rationality.

The point is not to assert that agents are purposeless and act randomly: zero-intelligence means that starting from individual behavior or rationality, macro phenomena cannot be predicted. In short, since rationality has no observable impact on market data, the rationality hypothesis may be superfluous.

This development of the macro-micro relationship, the true crux of econophysics, is not the only new element of this third phase, which is also distinguished by the prominence given to other fields. In fact, econophysics begins to be widely interested in the real economy in production and enterprises. A broad econophysics approach to production (the so-called “classical econophysics”) has been proposed by Cockshott, Cottrell, Michaelson, Wright and Yakovenko, in a volume published in 2009, *Classical Econophysics* [33]. The title is explained in the following terms by the authors of the first part of the book: classical physics, from Galileo to Bohr plus classical economics, from Smith to Marx. We could say: econophysics devoted to work and energy on the one hand, and classical political economy focused on economic development on the other. The goal is actually even more ambitious than building an econophysics from classical physics and economics: the authors identify several categories that could unify the two disciplines, physics and economics.

Classical econophysics is close to the field of political economy, as highlighted by the treatment of “value”—a concept forgotten by neoclassical economics, and reinterpreted here based on “simulation data, empirical data, and statistical mechanics arguments” (p. 3) [35]. There is much interplay between physics and economics: from energy/value and energy/utility parallelism to fluid/monetary flow, to the common ground of technological innovation [36]. However, it is the relationship between thermodynamics and economics (hardly a new topic), with its burden of “entropy” and information, that remains at the heart of any econophysics view of production. In a nutshell, the point is: thermodynamics implies the conservation of energy, a principle that so far has not been confirmed in economic processes.

### 5.4. Phase IV—Emergent Properties

No concept is abandoned, but in the fourth phase (2013–2017) the frontiers of econophysics seem to be expanding, as highlighted by the repeated use of “complex systems.” A “complex system” is “a system with a large number of mutually interacting parts, often open to their environment, that self-organize their internal structure and dynamics with new and sometimes surprising ‘emergent’ macroscopic properties” (p. 3196) [37]. The macroprospective is anchored in the idea that particles have “emergent properties,” i.e., that [emergent properties] produce effects that are only visible at the macro level. Emergent properties originate from self-organization due to nonlinear interactions between humans or heterogeneous agents. It should be recognized that statistical econophysics does not provide a clear formulation for the occurrence of emergent properties. Econophysics looks at emergent properties because at the macro dimension. It is also interesting that physicists confess that they cannot predict the exact shape of these phenomena [38]: analysis of emergent properties requires tools other than those drawn from statistical physics.

The point is that the concept of “emergent property” was primarily devised by Keynes in economics, not physics, but has never been adequately developed in modern economics. Perhaps this is because, unlike econophysicists, economists base their reasoning on a movement from micro-level structures to complex global-level structures. Emergent properties involve phenomena that can only be observed at macro-level structures, where objects are irreducible to their components. They cannot be microfounded. Statistical physics states that it is not necessary to define the properties of particles or components. What matters are their effects at the macro level where the emergent properties are visible.

Emerging properties of systems are produced at the meso/macro level, the study of which requires new concepts: network is one of them. The occurrence of the words “network” peaked in 2014, after increasing considerably in 2012 and 2013 (weighing for the 0.2 per cent of horizontal axis and 0.7 percent of vertical one). The network shaped a real trend in econophysics studies during that period. The study of aggregates of indistinct particles/agents, followed by attention to the self-organizing capabilities of these particles/agents, paved the way for connections between agents and/or sets of agents, and their ability to build networks in financial and economic contexts. Graph theories provide the mathematical basis for the scientific description of networks. In 2014, Slanina wrote: “Numerous interdependences we find in society can be expressed in terms of a collection of networks, each of them mapping a certain aspect of pairwise interactions among humans or human collectives, or even products of human activities” (p. 222) [39].

Bargigli and Tedeschi wrote: “Network theory deals with the structure of interaction within a multiagent system. Consequently, it is naturally interested in the statistical equilibrium of these systems […] Following this path, we come close to the idea […] of reconstructing macroeconomics under the theoretical framework of statistical physics and combinatorial stochastic processes” (p. 2) [40]. The need to understand interactions at the meso and macro level fostered the growth of network analysis, which gradually became one of the foundations of “macroeconophysics”. However, it is precisely the increased attention that has fostered its consolidation as an independent research area with respect to econophysics, as highlighted by the distribution of articles in *Physica A* in which “econophysics” and “network analysis” identify two distinct subareas.

One aspect of the explosion of attention to “network analysis” is a further broadening of financial market studies, as shown by the peaks in Figure 1. De Area Leão Pereira et al. (p. 258) [41] gave the first reason for this when they wrote: “The use of complex networks in financial markets has enabled a new view, mainly to measure the financial interaction between stock exchanges, assets, banks or companies. In this case, the nodes are usually assets, banks or countries.” Complex networks add the interdependence of markets as a necessary condition for studying the fragility of financial systems. As with emergent properties and other topics, “network analysis” is brought back into the realm of statistical physics.

The segment “complex networks”, which occurred more often than “complex system” in 2017, does not only refer to the financial world. It also includes production and business networks, reinforcing econophysics in the direction of the real economy as well as the financial economy. Econophysics was born with financial markets, and finance remains at the heart of this discipline. The question, however, is which econophysics is best suited to investigate production.

After rediscussing the temporal dimension, the other dimension to consider is space: terms such as “international network” and “macroeconomics” testify to a particular and gradual shift to great spaces. In 2016, Paul Ormerod wrote, “There is a great opportunity for econophysicists in the area of macroeconomics. Mainstream [DSGE] models are felt to be unsatisfactory, both by policy-makers and by mainstream economists” (p. 3288) [42]. Reference to communities of production networks [43] shifts econophysics to a spatial dimension that inevitably draws attention to the multiple connections that link productive or financial vertices at the international level. At these vertices we can find institutions, firms, industries, central banks, as well as agents. Econophysics is thus enriched by macro-econophysics, an important new field that opens up possibilities and raises challenges.

Consistently with these macro developments, the fourth is also the subarea comprising topics such as monetary and banking relationships, an operational field that, until then, has played a marginal role in econophysics [44].

### 5.5. Phase IV—Cryptophysics

Looking at Figure 1, one may wonder if there is a discontinuity between the fifth phase, which covers the years 2018 to 2020, and the previous phase. Some of the topics reported—bitcoin, cryptocurrency and sentiment analysis—seem far removed from the tradition of econophysics. “Sentiment” weighs 1 percent of the horizontal axis. A few trends can be detected.

First, it seems that econophysics is looking increasingly at macroeconomics and the real economy. The area is populated with words like “worker”, “factory,” “productivity” and “profit”. It is decidedly interesting that a word as “profit” contributes in determining the axes (0.3 and 0.1 respectively). Are we facing a definitive shift to the real phenomena of the economy? Only in part. However, an “economic” strand of research seems to be consolidating, covering the firm [45], the price of crude oil (analyzed both financially and as a commodity) [46], capital income [47] and economic policy [48]. “Oil” weighs 1.1 percent in the horizontal axis and 0.8 percent in the vertical axis. “Crude oil” for 0.6 and 0.2 respectively. Innovations in methods and analytical tools are anchored in content with increasing areas of overlap with economics.

One may wonder why the attention to productive and industrial or economic-social issues does not explode, even if a growth of interest in real economy is undoubted. The doubt that arises is that econophysics remains tied to concepts and tools that, in a sense, prevent a decisive enlargement of the research area.

Second, the CURR* lexical cluster emerges strongly here. The word “bitcoin” alone contributes 3 percent of the horizontal axis and 2.3 percent of the vertical axis. Not only bitcoin, but “cryptocurrency” contributes 0.4 and 0.6 in structuring the axes and “cryptocurrencies” 0.3 and 0.5 respectively. Bitcoins are analyzed as financial assets and means of exchange [49,50,51,52]. According to this qualitative analysis, FIN* and CURR* converge. The same “gold” matters for 0.3 and 0.6 of horizontal and vertical axis. From the centrality of “option pricing” in the early years, to the relevance of “cryptocurrency pricing” in recent years [53,54].

Third, the consolidation of the “quantum walk” as a development of the now historical “random walk” opens new fields of application that, at least from a lexical point of view, seem to change econophysics. The “quantum communication” leads to “quantum cryptographic protocols” [55] (semi-quantum key distribution, among others), which seem to open to further enrichment of econophysics. In terms of content, the four mentioned above and CURR* in particular seem to be sufficient to contain also these developments that pertain mainly to the instrumental aspect.

All that being said, FIN* and DIST* remain the two central topics in econophysics [56,57,58,59], gradually joined by PROD*, CURR* and NETW*. The core of econophysics does not change, although the focus on specific phenomena induces continuous enlargement of the toolbox.

To conclude, Figure 3 shows that in the years 2018–2020 our word cloud undergoes a dilation and the distance between the words/segments that characterize those years and the core lexicon of econophysics tends to increase, as evidenced by the widespread presence at this stage of words/segments that weigh in the structuring of the scatterplot. This means that the use of the new words/segments has less need of the lexical apparatus typical of statistical mechanics, which is located around the origin of the axes, than was previously the case when new terms were introduced.

## 6. Concluding Remarks

A first conclusion of this lexical investigation is that econophysics can certainly be included among the attempts of synthesis between natural scientific language and economic and social language. Figure 3 speaks to both worlds, natural and social.

The words/segments legacy of statistical physics lie in the center (centroid) of the Figure 3, which, however, tells us that there is a dialectical relationship between this core of words/segments and words that over the years take over the scene, conditioning in some way the scientific debate within econophysics. It happened with the word “agents” in the first phase; with the word “network” but also “crude oil” in the fourth phase; with “bitcoin,” “cryptocurrency,” “sentiment,” “gold” in the fifth and most recent phase. This is how the lexicon of econophysics evolves.

Does this dialectical process affect the propensity of econophysicists to seek “power laws” and invariant laws? Jovanovic and Schinckus stated: “The implicit disciplinary assumptions that econophysicists have regarding the identification of statistical laws come from the hypothesis of the universality of power laws. To put it in other words, econophysics inductively expects to identify a power law” (p. 37) [12]. However, the finding that linguistic variability increases in the last stages considered here (the fourth and fifth), together with the increased frequency of the NONST* lexical cluster from 2017 onwards (see Figure 2), leads us to conclude that the search for invariant laws is a fact that is far from being definitively established in econophysics.

The discipline seems to evolve on the basis of a different and less obvious point of attraction than “power law”: the dialectical process that arises from the application and questioning of concepts and methods often drawn from statistical physics. The application of a complex, non-reductionist approach to observed phenomena seems to lead to the continued use of dialectical, if not contrasting concepts, as suggested by the oscillating values of the STAT* and NONST* time series.

In Figure 2 there is no bifurcation. The development of econophysics seems to depend on the intertwining and contamination between these conflicting concepts, rather than on the assertion of one orientation or the other, STAT* or NONST*.

The process of consolidation and enlargement of lexical clusters on the one hand reinvigorates the debate on stationarity and non-stationarity, in short, on the application of statistical physics to economic and social relations, on the other, it is the product of that debate.

Figure 3 and the lexical analysis of these twenty-two years show that the evolution of econophysics does not depend so much on the consolidation of certain principles, approaches or visions as on their continuous questioning and enrichment with other contents and areas.

To conclude, the effectiveness of “power law” does not seem to be a consequence of its universality, but rather of its non-dogmatic use which requires continuous verification. A conclusion that also seems relevant to the other pillars of econophysics—“scale invariance,” “multifractality,” and so on—and to the overall application of statistical physics to the social sciences.

## Figures and Tables

**Figure 1 entropy-23-00944-f001:**
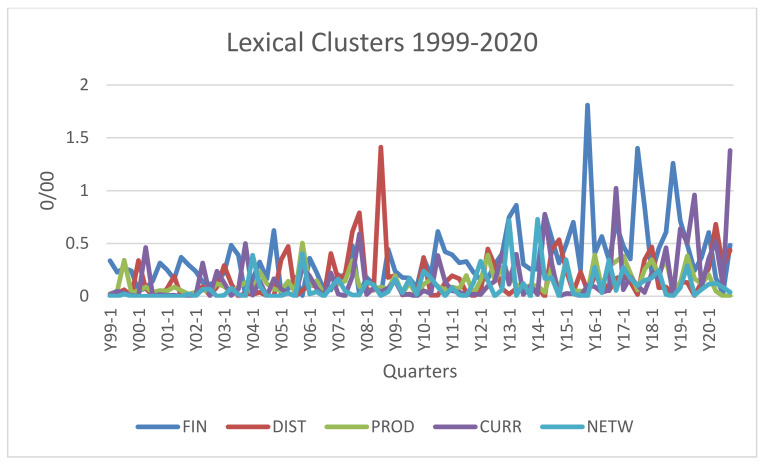
Econophysics main lexical clusters from 1999 to 2020.

**Figure 2 entropy-23-00944-f002:**
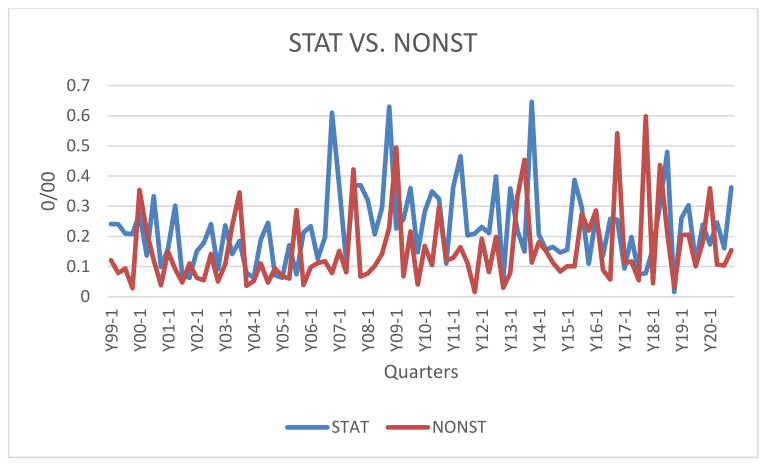
Lexical clusters on STAT* and NONST* approaches from 1999 to 2020.

**Figure 3 entropy-23-00944-f003:**
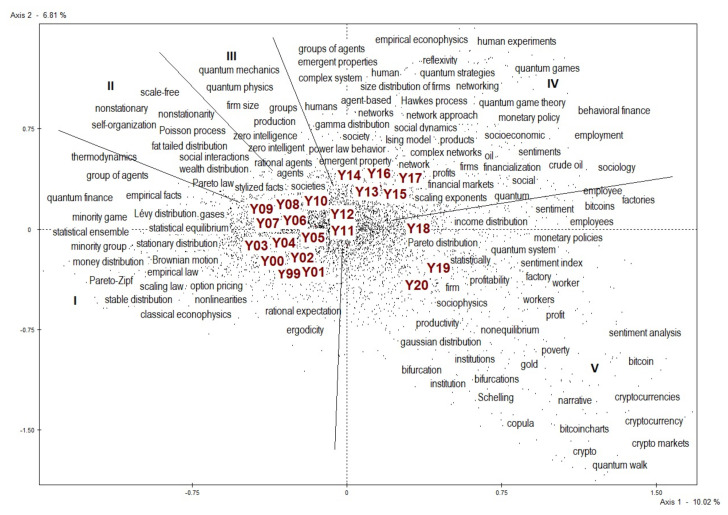
The vocabulary of econophysics between 1999 and 2020.

## Data Availability

Data sharing not applicable.

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
