# Peer review of "Plotting the Words of Econophysics"

_entropy, 2021, doi:10.3390/e23080944_

Round 1

Reviewer 1 Report

It is a very interesting paper on the study of econophysics. I recommend it to be published at the Entropy.

Author Response

Dear Reviewer,

Many thanks for the flattering evaluation of my manuscript.

Sincerely yours

Gianfranco Tusset

Reviewer 2 Report

The paper investigates an interesting topic about the research directions and their evolution of Econophysics. The paper fits well the scope of this Special Issue.

My main concern is about the data. The author wrote: "For construction of the matrix and, thus, the scatterplot, 510 econophysics articles published between the years 1999 and 2020 were used, mainly in Physica A (287 or 56%), partly in The European Physical Journal B (165, 32%), and a minority (58, 12%) from other journals (Physical Review E, Contemporary Physics and few others). In the article only the words were used: therefore, the mathematical or statistical content is not taken into account." A few questions rise.

How are the 510 econophysics papers determined or selected? What are the criteria? Obviously, econophysics is no more an emerging field because it has developed for about 30 years since Mantegna's work in 1991. There are much more econophysics papers published. For instance, in a review paper "Multifractal analysis of financial markets: a review" [Reports on Progress in Physics 82 (12), 125901 (2019)], 1101 references are listed and there are "more than 510" papers for sure.

How are the journals determined or selected? The author listed a few physics journals and even them are not a complete list. Actually, there are also many econophysics papers in non-physics journals, expecially in finance journals.

Therefore, a more general question is the "definition" of econophysics papers. I understand that it is almost impossible to give a operational definition and to retrieve all the econophysics paper for analysis. Nevertheless, the author should make the situation clearer about the limitation of the data and the possible bias of the results.

For Ref. [1], maybe it is better to cite "ETTORE MAJORANA, The value of statistical laws in physics and social sciences, Quantitative Finance, Vol. 5, No. 2, April 2005, 133–140", which was translated by R.N. Mantegna.

Author Response

Dear Reviewer, 
Thanks for your comments.
I have tried to clarify the criteria followed to identify the published articles that formed the basis of the linguistic corpus (p. 3). 
I am fully aware of the partiality of the number of articles covered, but this is what I could do considering that each article must be transformed into txt format, cleaned, corrected (the transformation of some words appears incorrect) and prepared for analysis. It is my intention to expand the size of the corpus by including articles from more journals and more sources. This is a goal for future research. 
To maintain some correspondence between the content of the downloaded articles and the articles published in the two journals considered, I grouped the published articles consistently with the identified lexical clusters and selected a portion of them in order to maintain proportionality between what had been published and what I would use.

Cordially,
Gianfranco Tusset

Reviewer 3 Report

See attachment.

Author Response

Dear Reviewer,

Thanks for your comments.

Below is a list of some of the changes made to address your requests:

- An appendix was added with the most significant values of the Granger causality test on lexical clusters;

- I tried to better explain criteria of construction and method of analysis of the linguistic corpus;

- Guidance has been added for the interpretation of Figure 3;

- Meaningful data about the relevance/placement of words have been inserted;

- I better specified objectives and outcomes of the paper;

- Finally, other minor corrections have been made.

However, I did not turn the manuscript into a fully technical paper as you requested.

In truth, it was not my intention to propose a technical article, but a lexical analysis where the statistical part is functional to a qualitative analysis.

Why did I submit a manuscript with these characteristics?

The first reason is that participation in the special issue had been suggested to me by editors familiar with my areas of research.

Second, but no less important, I interpreted the invitation of the special issue as also addressing work that would speak to a non-specialist audience, outside the traditional boundaries of econohysics.

For these reasons I decided to submit it, without any claim to contribute to the technical advancement of the discipline.

Kind regards

Gianfranco Tusset

Reviewer 4 Report

This article applied text mining to show the progress of the econophysics. We can observe the stream of the research in econophysics. I accept the article to publish.

Author Response

Dear Reviewer,

Many thanks for the encouraging evaluation of my manuscript.

Kind regards

Gianfranco Tusset

Round 2

Reviewer 2 Report

I did not find point-by-point reply. I cannot find my first report on the system neither. The author should copy my report and then reply.

Round 3

Reviewer 2 Report

Now the reply of the author is much clearer. I do not have further comments or suggestions.